# Uridine Diphosphate Glucose (UDP-G) Activates Oxidative Stress and Respiratory Burst in Isolated Neutrophils

**DOI:** 10.3390/ph16101501

**Published:** 2023-10-21

**Authors:** Fabiana Lairion, Claudio Carbia, Iris Maribel Chiesa, Christian Saporito-Magriña, Natalia Borda, Alberto Lazarowski, Marisa Gabriela Repetto

**Affiliations:** 1Cátedra de Química General e Inorgánica, Departamento de Ciencias Químicas, Facultad de Farmacia y Bioquímica, Universidad de Buenos Aires, Buenos Aires 1113AAD, Argentina; flairion@ffyb.uba.ar (F.L.); irischiesa87@gmail.com (I.M.C.); csaporito@ffyb.uba.ar (C.S.-M.); 2Instituto de Bioquímica y Medicina Molecular Prof. Alberto Boveris, Consejo Nacional de Investigaciones Científicas y Tecnológicas (IBIMOL, UBA-CONICET), Buenos Aires 1113AAD, Argentina; 3Cátedra de Bioquímica Clínica II-Área Hematología, Departamento de Bioquímica Clínica, Facultad de Farmacia y Bioquímica, Instituto de Fisiopatología y Bioquímica Clínica (INFIBIOC), Universidad de Buenos Aires, Buenos Aires 1113AAD, Argentina; ccarbia@ffyb.uba.ar (C.C.); nataliasborda@gmail.com (N.B.)

**Keywords:** neutrophils, PMN, UDP-glucose, P2Y_14_-R, oxidative stress, free radicals, respiratory burst, NADPH oxidase

## Abstract

The extracellular purinergic agonist uridine diphosphate glucose (UDP-G) activates chemotaxis of human neutrophils (PMN) and the recruitment of PMN at the lung level, via P2Y14 purinergic receptor signaling. This effect is similar to the activation of PMN with N-formyl-methionyl-leucyl-phenylalanine (fMLP), a mechanism that also triggers the production of superoxide anion and hydrogen peroxide via the NADPH oxidase system. However, the effects of UDP-G on this system have not been studied. Defects in the intracellular phagocyte respiratory burst (RB) cause recurrent infections, immunodeficiency, and chronic and severe diseases in affected patients, often with sepsis and hypoxia. The extracellular activation of PMN by UDP-G could affect the RB and oxidative stress (OS) in situations of inflammation, infection and/or sepsis. The association of PMNs activation by UDP-G with OS and RB was studied. OS was evaluated by measuring spontaneous chemiluminescence (CL) of PMNs with a scintillation photon counter, and RB by measuring oxygen consumption with an oxygen Clark electrode at 37 °C, in non-stimulated cells and after activation (15 min) with lipopolysaccharides (LPS, 2 µg/mL), phorbol myristate acetate (PMA, 20 ng/mL), or UDP-G (100 μM). The stimulation index (SI) was calculated in order to establish the activation effect of the three agonists. After stimulation with LPS or PMA, the activated PMNs (0.1 × 10^6^ cells/mL) showed an increase in CL (35%, *p* < 0.05 and 56%, *p* < 0.01, SI of 1.56 and 2.20, respectively). Contrariwise, the stimulation with UDP-G led to a decreased CL in a dose-dependent manner (60%, 25 μM, *p* < 0.05; 90%, 50–150 μM, *p* < 0.001). Nonetheless, despite the lack of oxidative damage, UDP-G triggered RB (SI 1.8) in a dose-dependent manner (38–50%, 100–200 μM, *p* < 0.0001). UDP-G is able to trigger NADPH oxidase activation in PMNs. Therefore, the prevention of OS and oxidative damage observed upon PMN stimulation with UDP-G indicates an antioxidant property of this molecule which is likely due to the activation of antioxidant defenses. Altogether, LPS and UDP-G have a synergistic effect, suggesting a key role in infection and/or sepsis.

## 1. Introduction

An extracellular physiological function of nucleotides as purines and their corresponding nucleosides was initially described as “purinergic signalling” by Geoffrey Burnstock in 1970 [1]. Adenosine triphosphate (ATP) and its degradation products are released from cells to activate purinergic receptors (PR) expressed on the surface of different types of cells, promoting a wide range of physiological processes [2,3].

Within the three families of PR described are the G protein-coupled metabotropic receptors, including P2Y_14_-R, which can be activated by uridine diphosphate glucose (UDP-G), or other UDP-sugars [4], with a relative affinity potency order of UDP-glucose > UDP-galactose > UDP-glucuronic acid > UDP-N-acetylglucosamine [5].

The P2Y_14_-R mRNA expression has been reported in several organs, such as the brain, lungs, immune or inflammatory cells, and hematopoietic stem cells. Interestingly, unlike other cell types (dendritic cells, T lymphocytes, bone marrow cells, and astrocytes) [6,7,8], polymorphonuclear cells (PMNs) have a very high transcript load (mRNA) of P2Y_14_-R, being 14,000 times greater than in the aforementioned cell types [7].

Very few studies have described the specific action of extracellular UDP-G signaling on PMNs as the neutrophil recruitment in the lung, and the promotion of chemotaxis (CTX) of freshly isolated human neutrophils [9,10]. Furthermore, a markedly increased P2Y_14_-R expression during an induced neutrophilic differentiation from HL60 human promyelocytic leukemia cells was described, as compared with HL60 naive cells that do not express the P2Y_14_-R [11]. 

Among the different cell types capable of secreting UDP-G are the goblet epithelial cells present in the bronchial epithelium [5]. Furthermore, these cells generate the secretion of UDP-G coupled to mucin secretion, as demonstrated using goblet-like Calu-3 cells [12], as well as human bronchial epithelial cell cultures that were induced to develop goblet (mucous) cell metaplasia [13].

In previous work, it has been shown that UDP-G promotes CTX of normal human neutrophil PMNs, and such activation was most pronounced in PMNs preincubated with serum from patients with rheumatoid arthritis (RA) and high levels of TNF-α [14]. Inflammatory cells infiltrate the airways of pediatric patients with respiratory syncytial virus (RSV)-induced bronchiolitis, and approximately 80% of infiltrated cells are neutrophils [15]. Furthermore, neutrophil degranulation primes neutrophils for reactive oxygen species (ROS) generation by mobilizing nicotinamide adenine dinucleotide phosphate (NADPH) oxidase complex components to the plasma membrane [16], and the Influenza A virus and other infections cause the generation of ROS, mainly superoxide anion (O_2_**^.^**), hydrogen peroxide (H_2_O_2_), and singlet oxygen (^1^O_2_) in neutrophils [17].

An oxidative stress (OS) situation implies reversibility of the increase in biochemical oxidants or decrease in antioxidants, in a progressive and continuous process, including adaptive response, by the synthesis and activation of antioxidants and the upregulation of enzymes or signaling mechanisms to control the redox homeostasis in biological systems [18].

Oxidative damage is characterized by increases in oxidized biomolecules, as a consequence of lipid and protein oxidation, that implies irreversibility of cell toxicity and death [19]. Currently, a new concept of OS has emerged. Oxidative damage is defined as distress and it should be distinguished from eustress, including physiological stress, signaling, and hormetic actions. These processes are dependent on the concentration of oxidant species, where lower concentration of oxidants induce beneficial signaling effects (eustress) whereas higher levels lead to cellular damage (distress) [20].

Spontaneous chemiluminescence (CL) determination is a non-invasive, non-destructive indirect assay based in measuring the emission from the electronic transition from an excited state to a basal state of active species as ^1^O_2_ and peroxyl radicals (ROO^.^), which are produced in the phospholipid peroxidation chain reaction. The electronically excited state of ^1^O_2_ quickly returns to the basal triplet with the emission of a photon. The number of photons emitted is dependent on the steady-state concentration of ^1^O_2_ and ROO^.^, both in physiological and pathological conditions. The photon emission is proportional to the ^1^O_2_ excited state, which is an indicator of the oxidative damage [18,19,20].

PMNs are the most abundant circulating white blood cells in humans (typically between 40–75%) producing O_2_**^.^** and H_2_O_2_ via the NADPH oxidase complex. As both the first line of innate defense and effectors of adaptive immunity, PMNs play crucial roles in immune defense against bacterial, fungal, and even viral infections [21].

Moreover, previous studies have demonstrated that in inflammation processes with abundant neutrophils in the human alveoli on day 3 post-infection with SARS-coronavirus (SARS-CoV) and at day 5 post-infection, more inflammatory cells, including foamy neutrophils, were present in the alveoli [22]. The COVID-19 coronavirus pandemic is a challenge to global public health, and very few data indicating PMN infiltration into the lungs of COVID-19 infected cases has been reported [23]. Furthermore, abundant intra-alveolar neutrophilic infiltration and the formation of neutrophil extracellular traps (NETs) were consistent with superimposed bacterial bronchopneumonia [22,24]. This mechanism may be considered as an aggravating factor in the setting of respiratory failure caused by coronaviruses.

The objective of this research was to demonstrate if one of the biochemical mechanisms involved in the activation of PMNs by UDP-G is OS. The present in vitro study shows that the extracellular UDP-G induces the oxidative stress and RB of PMN.

The scope of the article suggests a key role for UDP-G in the PMN response to infection and/or sepsis.

## 2. Results

In order to measure the activation of the oxidative response and lipid peroxidation, it was necessary to first standardize the optimal cell concentration. We used spontaneous sensitive methodologies, and for this reason optimization of the experimental variables, such as PMN concentration, is required before their use.

### 2.1. Standardization of Optimal PMN Concentration

#### 2.1.1. Oxidative Stress Measured as Spontaneous Chemiluminescence

The standardization of optimal PMN concentration for the OS evaluation measured by CL indicated that stimulations with LPS (*p* < 0.05) or PMA (*p* < 0.01) induced higher CL as compared with the naive PMNs (control) (Figure 1). However, the results of CL after stimulation with LPS and PMA were similar to the control when the assays were developed at concentrations of PMNs higher than 0.5 × 10^6^ PMN/mL (Figure 1), but, when assay was carried out with concentrations ranging from 0.015 to 0.15 × 10^6^ PMN/mL (Figure 1), a higher sensitivity was observed.

In non-stimulated PMNs, the spontaneous CL was 8.65 ± 1.99 cps/cm^2^, and CL increased by 35% (*p* < 0.05) and 56% (*p* < 0.01) (Figure 2) with LPS and PMA, respectively.

Similarly, UDP-G led to an increased CL at the same low concentrations of PMNs mentioned above (*p* < 0.001) (Figure 3).

However, under concentrations higher than 0.8 × 10^6^ PMN/mL, UDP-G appears to inhibit the CL (*p* < 0.001, Figure 4), which was not observed in the case of LPS or PMA (Figure 2).

#### 2.1.2. Respiratory Burst Measured as Oxygen Consumption

The optimal PMN concentration to evaluate O_2_ consumption after stimulation with LPS and PMA was between 0.1 and 0.5 × 10^6^ PMN/mL. At a PMN concentration of 0.5 × 10^6^ PMN/mL, PMA showed greater stimulation (40%, *p* < 0.05) than LPS (15%, *p* < 0.001) with respect to the control (Table 1). In activated PMNs with PMA the optimal PMN concentration was 0.1 × 10^6^ cells/mL, showing a significant increase of 100% (*p* < 0.001).

Similarly, when stimulation with UDP-G was tested, an increase in O_2_ consumption was also observed (*p* < 0.001). However, again, at higher concentrations of PMNs, O_2_ consumption decreased below control values (*p* < 0.001, Figure 5), which was not observed with LPS or PMA). At two different low PMN concentrations (0.1 and 0.5 × 10^6^ PMN/mL) under basal conditions (Control), O_2_ consumption were 7.2 ± 1.3 and 1.8 ± 1.2 nmol O_2_/min/10^6^ PMN, respectively. At these both low PMN concentrations, UDP-G increased RB 100% (*p* < 0.001) or 80%, respectively (*p* < 0.01). At higher PMN concentrations (1.0 × 10^6^ PMN/mL), O_2_ consumption with UDP-G was lower than the control (*p* < 0.001, Figure 5).

When compared with LPS and PMA, the stimulating effect of the UDP-G on the O_2_ consumption of the PMNs (0.1 × 10^6^ PMN/ mL and 0.5 × 10^6^ PMN/ mL) was higher compared to that produced by PMA (*p* < 0.001) and, strikingly, the LPS stimulus was not significantly different from the control at these concentrations of PMNs (Table 2).

An 80% increase in RB is induced by UDP-G at lower concentrations of PMNs (0.1 × 10^6^ PMN/mL) (*p* < 0.001). This increase rises to 100% at a concentration of 0.5 × 10^6^ PMN/mL (*p* < 0.001). No effect was observed with LPS.

### 2.2. PMN Activation Effect of Agonists: Stimulation Index

The stimulation index (SI) is an indicator of the inflammatory and oxidative response of PMNs exposed to different agonists and allows comparison of the activating effect. When working with a very low PMN concentration of 0.02 × 10^6^ PMN/mL, no significant differences in the SI were observed with any of the agonists studied individually. However, O_2_ consumption was observed to reach a 26% increase (*p* < 0.05) when the PMNs (0.02 × 10^6^ PMN/mL) were sequentially incubated with LPS plus UDP-G (Table 3) with respect to the control cells (55 ± 10 nmol O_2_/min/ 10^6^ cells), with an SI of 1.3, indicating that LPS and UDP-G have synergic effects on the activation of PMNs. The synergism in respiratory burst (RB) measured as O_2_ consumption in PMNs sequentially incubated with LPS plus UDP-G, at a concentration of 0.02 × 10^6^ PMN/mL, as compared with a control (69 ± 4 nmol O_2_/min/10^6^ PMN vs. 55 ± 10 nmol O_2_/min/10^6^ PMN, *p* < 0.05) was observed and is shown in Table 3.

In summary, the optimal cell concentration to evaluate OS and RB in PMNs was 0.1 × 10^6^ PMN/mL. The SI for spontaneous CL (CL-SI) was 1.56 and 2.20, and the SI values for RB (RB-SI) were 1.2 and 1.4 with LPS or PMA, respectively, for 0.1 × 10^6^ cells/mL. When PMNs were stimulated with UDP-G (100 µM), the CL-SI was 1.6 and the RB-SI was equal to 1.8.

In Table 4, the SI values of CL and RB for LPS, PMA, and UDP-G on activated PMNs (0.1 × 10^6^ PMN/mL) are listed.

### 2.3. Effect of UDP-G on PMN Function: OS and RB

When UDP-G was used as stimuli on PMNs (0.1 × 10^6^ PMN/mL), CL decreased with increments of UDP-G concentration (60%, 25 µM, *p* < 0.05; 90%, 50–150 µM, *p* < 0.001) (Figure 6), and oxygen consumption was significantly increased with UDP-G concentration (C:88 ± 15 nmol O_2_/min/10^6^ PMN) (38%, 100 µM to 52% with 200 µM, *p* < 0.0001) (Figure 7).

## 3. Discussion

This is the first study that allows the evaluation of the inflammatory response and OS of living cells, ex vivo, by measuring the oxygen consumption and CL in PMNs. This research provides optimal working conditions for the determination of OS in these cells using sensitive techniques. This work shows that this methodology can be applied to evaluate the functionality of living cells in physiological and/or pathological conditions, as well as the effect of stimulating or inhibitory agents of the inflammatory, oxidant, or antioxidant response. In this study, it was observed that, at the extracellular level, UDP-G exerts a dual effect on exposed PMNs, an increase in RB and ROS production, and oxidative damage to membrane lipids at lower PMN and UDP-G concentrations. However, UDP-G concentrations greater than those that promote CTX and inflammatory response (100 µM) increase the RB but decrease oxidative damage, indicating that the ROS generated are destined for the inflammatory and defense response of the PMNs. However, the main limitations of this study are that the cells are required to be kept alive throughout the experiment, they have to be isolated from the blood on the day of sampling, and both oxygen consumption and CL must be measured immediately.

UDP-G is not only a key component of intracytoplasmic glycosylation reactions at the endoplasmic reticulum and Golgi apparatus levels, but it is also a molecule with extracellular signaling activity acting as agonists on the P2Y_14_ receptor (P2Y_14_R) [25].

It was described that extracellular purinergic nucleotides and nucleosides can regulate the mucociliary clearance activity in the airways, mediated mainly by the P2Y_2_R and the A_2B_R [14,15,26].

Additionally, it was also demonstrated that UDP-glucose can activate P2Y_14_-R and induce the release of the potent neutrophil chemoattractant interleukin 8 (IL-8) [6].

In vitro, UDP-G was able to induce P2Y_14_R-mediated chemotaxis in freshly isolated human PMNs, or neutrophil-like HL60 cells, in a gradient-dependent manner [10,11]. Furthermore, the administration of UDP-glucose into mouse lungs results in neutrophil inflammation [11].

The OS and RB of PMNs are directly related to ROS production as a powerful antimicrobial weapon and the major component of the innate immune defense against bacterial and fungal infections [9]. ROS must be released where they are needed into the intracytoplasmic phagosome to kill bacteria, as well as to the extracellular environment at the site of infection for neutrophil extracellular traps (NETS) induction, and they stimulate the production of the pro-inflammatory cytokines [27,28].

In the present study, the OS and RB of PMNs were evaluated by spontaneous CL and O_2_ consumption, respectively. Here, we demonstrated that UDP-G increased these activities when experiments are developed using low concentrations of PMNs, and the same effect that was observed for LPS and PMA activators; conversely, exposure of higher PMN concentrations to UDP-G showed results clearly below the control values (Figure 4 and Figure 5).

These results suggest a greater sensitivity of PMNs to the stimuli received with high concentrations of the agonists, and in consequence, progressive infiltration of PMNs into tissues (e.g., lung), which would indicate that as more PMNs enter the tissue, the lower its ability to activate PMNs. Furthermore, the inverse result observed with UDP-G and high concentrations of PMNs suggests that, under these conditions, UDP-G would cause inactivation or immunoparalysis, or that the amount of UDP is not enough to activate such a high PMN load.

Additionally, at very low PMN concentrations, in the range of 1 × 10^4^ cells/mL, none of the three agonists stimulated the PMNs. However, the sequential administration of LPS plus UDP-G to these low concentrations of PMNs activated the consumption of O_2_ (Table 3). This effect suggests that a synergistic action between LPS and UDP-G could be present under clinical conditions of infection; LPS induces the release of lactoferrin, a glycoprotein with high iron chelating capacity present in the secondary granules of PMNs.

We propose that in the clinical setting, UDP-G could have a dual effect on the PMN priming, under different inflammatory or infective conditions. It is needed to obtain a better understanding of PMN behavior under different conditions, where their activation or disability can play a critical role as a tug-of-war in the balance between their protective and detrimental effects [22]. Furthermore, the ability of immune cells to infiltrate hypoxic tissues is a key feature for their protective role. The infiltration by neutrophils to hypoxic tissue as infarcted heart was reported very early, in the late 19th century [29]. Neutrophils are the first migrating cells to go to injured hypoxic tissues and, under this conditions, neutrophils prolong their survival, increase both their glycolytic and pentose phosphate pathway flux, and also upregulate the influx of extracellular glucose and glycogen storage [30].

The results of this research’s experiments indicate that UDP-G appears to have a dual behavior. Low cell concentrations induce PMN activation and, as a cellular response, the NADPH oxidase complex is activated, increasing extramitochondrial oxygen consumption in the PMN membrane. Consequently, the release of O_2_**^.^** and H_2_O_2_ increases, leading to the generation of hydroxyl radicals, initiating lipid peroxidation process, and inducing organic peroxides accumulation and release of free radicals, such as ^1^O_2_. Oxidative damage then occurs to phospholipids and proteins of the PMN membrane, with a consequent increase in CL. If the PMN concentration is high, UDP-G cannot protect cells from oxidative damage generated by the release of ROS in activated PMNs, with increases in RB, lipid peroxidation, and oxidative damage.

The determination of spontaneous CL is based on the measurement of the emission associated with the transition from the excited state to the ground level of the active species, such as ^1^O_2_. This occurs in the termination stages of the chain reaction of phospholipid peroxidation. The emission of photons is proportional to the steady state of ^1^O_2_, which in turn indicates oxidative damage by phospholipid peroxidation in vivo in the final stage of the chain reaction. UDP-G decreased spontaneous CL in PMNs, indicating that UDP-G may act as an antioxidant compound by preventing lipid peroxidation and OS in a dose-dependent manner.

When isolated PMNs were exposed to extracellular UDP-G, at a low UDP-G concentration, the CL in stimulated PMNs is lower than in non-stimulated PMNs. The decreasing CL rate was higher at low UDP-G concentrations until 50 µM, but the rate of light emission was reduced to zero after increasing the UDP-G concentration, reaching a stationary state of oxidized phospholipids, lipid peroxides, and ^1^O_2_ in the range of 100–200 µM UDP-G. These observations suggest that at higher UDP-G concentrations, the ROS generation rate and biomolecules oxidation rates are equal to the detoxification of oxidized species by the action of antioxidants.

Stimulation of PMNs induces RB and an inflammatory response. The effect of UDP-G on PMN activation and RB may be associated with increased production of O_2_**^.^** by the activation of the NADPH oxidase complex and can be measured by determining the extramitochondrial oxygen consumption of isolated PMNs with the Clark-type oxygen electrode. This oxidase-active multiprotein complex is normally inactive in quiescent cells, but, in response to stimulation, its components rapidly assemble on the cell membrane. The enzyme is activated and catalyzes the reduction of oxygen to form O_2_**^.^** and ROS via NADPH, such as H_2_O_2_ and hydroxyl radicals. UDP-G triggered the RB and NADPH oxidase activity in a dose-dependent manner.

Future experiments should confirm the decreased values of O_2_ consumption and RB described here, when high concentrations of PMNs are exposed to UDP-G, for a better understanding of the role of UDP-G not only in normoxia, but also under hypoxic conditions.

## 4. Materials and Methods

### 4.1. Isolation of Human PMNs and Cell Viability

This study was carried out on PMNs isolated from fresh blood samples anticoagulated with EDTA and obtained for the completion of the complete blood count in the Hematology Laboratory of the Department of Clinical Biochemistry of the Hospital de Clínicas José de San Martín, and/or voluntary blood donors of the Hospital, with normal results in hematic parameters. The investigation does not imply any study related to possible clinical–pathological situations; therefore, few samples are required and were selected for their normal results and without any identity link with the patient or donor. The venous blood samples have been obtained from subjects who have completed an informed consent form. The experimental protocol has been approved by the Clinical Research Ethics Committee, Secretariat of Science and Technology of the Faculty of Pharmacy and Biochemistry, University of Buenos Aires. Whole blood was collected in a microcentrifuge tube with EDTA and was then centrifuged at 1500× *g* for 10 min; the supernatant was discarded and PMNs were isolated using the density gradient separation technique and Ficoll–Hypaque centrifugation [31]. The obtained cell suspension was quantified using an automated hematological counter (Sysmex XT-1800i-Sysmex Corporation, Kobe, Japan). In all cell isolations, a PMN concentration >99% was reached, while mononuclear cells did not exceed 0.5%. Viability was tested with the vital staining Trypan blue solution (0.4%), and the cells suspension was adjusted to a final concentration of 2.5 × 10^6^ PMN/mL.

### 4.2. Stimuli and Activation of PMNs

Human PMN cells (1 × 10^4^–4 × 10^6^ cells) suspended in buffer phosphate solution (PBS: 138 mM NaCl, 2.7 mM KCl, 8.1 mM Na_2_HPO_4_, and 1.47 mM KH_2_PO_4_ buffer, pH 7.4) were preincubated for 15 min at 37 °C with phorbol 12-myristate 13-acetate (PMA, 20 ng/mL in ethanol), lipopolysaccharide (LPS, 2 µg/mL), or uridine diphosphate glucose (UDP-G) (0.1 mM) to activate the NADPH oxidase complex.

### 4.3. Standardization of Optimal PMN Concentration for OS and RB Measurement

Increased concentrations of PMNs in suspension were incubated with fixed concentrations of LPS and PMA and compared with fresh PMNs incubated with PBS. The PMN concentrations ranged from 0.015 to 1.0 × 10^6^ PMN/mL for OS evaluation, and from 0.01 to 1.0 × 10^6^ PMN/ mL for oxygen consumption. Later, assay of stimulations with UDP-G (100 µM), were developed under these standardized conditions. This UDP-G concentration was used because 100 µM was reported to promote CTX and the inflammatory response [14,15].

### 4.4. Biochemical Determinations of OS and RB

The OS was evaluated by measuring the spontaneous CL of cells with a scintillation photon counter, and RB was determined by measuring oxygen consumption with a Clark-type oxygen electrode at 37 °C, at basal condition (control, C) or after activation (15 min) with lipopolysaccharides (LPS, 2 µg/mL), phorbol myristate acetate (PMA, 20 ng/mL), or UDP-G (100 μM). These LPS and PMA concentrations are regularly used to trigger oxidative burst in PMNs [32], and, as was said previously, 100 µM of UDP-G was reported to promote CTX and inflammatory response [14,15].

#### 4.4.1. PMNs Spontaneous CL

Spontaneous CL determination in fresh PMNs was assayed using the procedure and device with a photon counter developed by Chance, Sies, and Boveris in 1979; the model in the Johnson Research Foundation of the University of Pennsylvania (Philadelphia, PA, USA) is used for the light emission reading [18,19,33]. The optimal PMN concentration required to evaluate the OS in PMN cells was carried out by assessing the CL achieved at different cell concentrations in non-stimulated cells and in the presence of LPS or PMA. Results are expressed in cps/10^6^ PMN (cps: counts per second; 1 cps corresponds to about 10^3^ photons per second).

#### 4.4.2. PMNs Oxygen Consumption

PMNs non-mitochondrial oxygen uptake was determined polarographically by high-resolution respirometry, using a Clark-type electrode (HasantechOxygraph System DW1) thermostated at 37 °C, with human PMN cells (1 × 10^4^× 10^6^ cells) in PBS supplemented with 0.9 mM CaCl_2_, 0.5 mM MgCl_2_, and 7.5 mM glucose (PBSG). For the assay, respiratory buffer (for 1 mL final volume) was placed in the electrode chamber. The chamber was covered. Substrates were added with a Hamilton syringe. The rate of oxygen uptake was calculated from the initial time course and expressed as nmol of oxygen/ min/ 10^6^ cells [18,31,34].

### 4.5. Activation Effect of Agonists: Stimulation Index

To establish the activation effect of the three agonists, the SI was calculated for each of the different treatments as the ratio (quotient) between the mean of the parameter evaluated in activated cells and non-activated cells.

### 4.6. Effect of UDP-G on PMN Function

To establish the effect of UDP-G on PMN function, increasing concentrations of UDP-G were evaluated (0–200 μM) and CL and RB values were determined at 25, 50, 100, 150, and 200 μM UDP-G. These concentrations were used because they are lower and higher than the UDP-G concentration (100 μM) that stimulates PMN response [14,15].

### 4.7. Statistical Analysis

Results are expressed as mean ± SD for normally distributed data. The Kolmogorov–Smirnov test was performed to evaluate the variables’ distribution. Differences between groups were tested using an unpaired Student’s *t*-test and a one-way analysis of variance (ANOVA), followed by the parametric Dunnet post hoc test. The software GraphPad InStat3 and GraphPhad Prism4 software (GraphPad Software Inc., San Diego, CA, USA) were used. Significances between the groups’ differences are indicated by the *p* value, where *p* < 0.05 was considered significant, *p* < 0.01 was considered very significant, and *p* < 0.001 was considered extremely significant.

## 5. Conclusions

In conclusion, extracellular UDP-G activates oxidative damage (distress) and RB at low PM concentrations. However, if the concentration of UDP-G increases, the oxidative damage to PMN membrane phospholipids decreases because OS become reversible (eustress), with an increase in RB and release of reactive oxygen molecules, such as O_2_**^.^** and H_2_O_2_, two known second messengers for cell signaling. A synergistic effect is observed after sequential administration of LPS and UDP-G, suggesting a key role of this combination in PMN activation during infectious and/or septic processes. In this regards, it was suggested that antagonism of the P2Y_14_ receptor could prevent the progression of COVID-19-induced systemic inflammation, which often leads to severe illness and death cases, as well as that P2Y14 receptor inhibition by its selective antagonist PPTN could limit neutrophil recruitment and NETosis, hence, limiting excessive formation of oxygen reactive species and proteolytic activation of the kallikrein–kinin system and subsequent bradykinin storm in the alveolar septa of COVID-19 patients [35].

## Figures and Tables

**Figure 1 pharmaceuticals-16-01501-f001:**
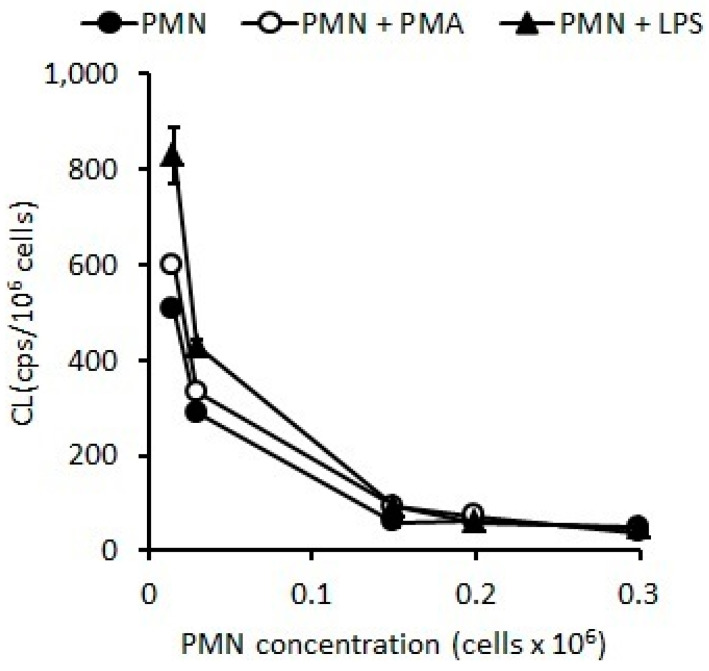
CL of PMNs stimulated with PMA (20 ng/mL) or LPS (2 ug/mL) for 15 min at 37 °C.

**Figure 2 pharmaceuticals-16-01501-f002:**
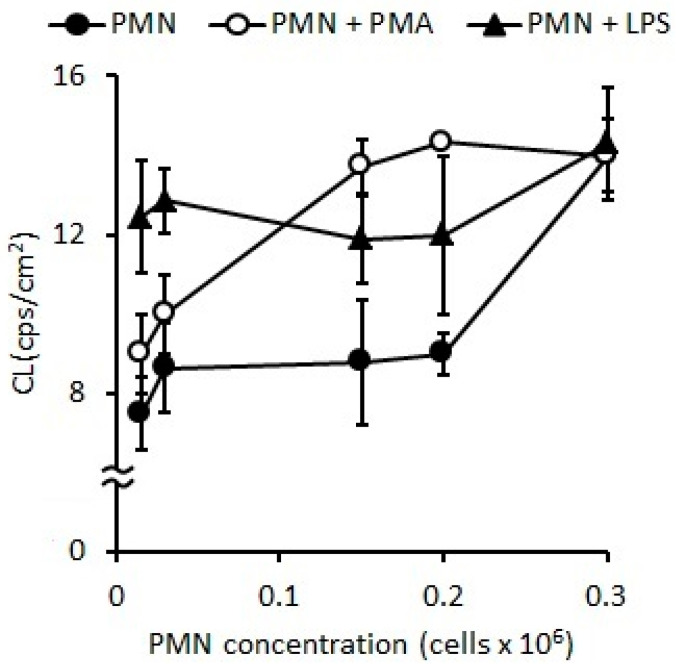
Changes in CL observed according to the concentration of PMNs stimulated with LPS (*p* < 0.05) or PMA (*p* < 0.01).

**Figure 3 pharmaceuticals-16-01501-f003:**
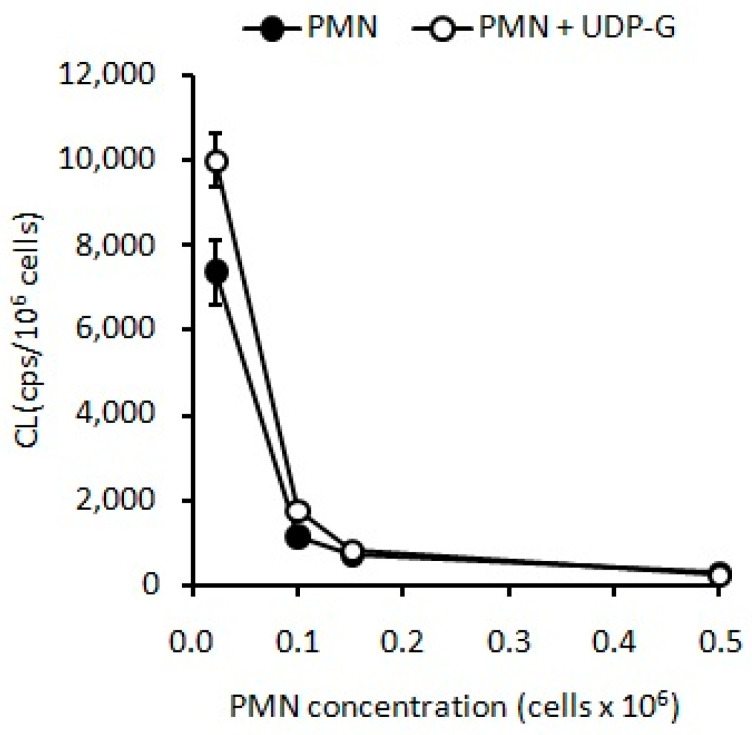
UDP-G induces a significant stimulation of CL at low PMN concentrations (*p* < 0.001).

**Figure 4 pharmaceuticals-16-01501-f004:**
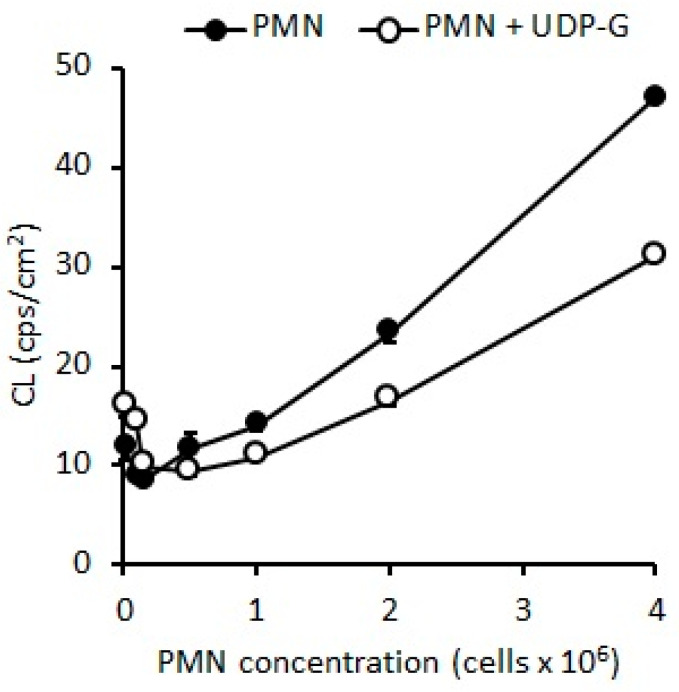
UDP-G appears to have a dual behavior, inducing CL activation at low PMN concentrations (0.015 × 10^6^ PMN/mL, *p* < 0.05, and 0.15 × 10^6^ PMN/mL, *p* < 0.01) but inhibiting CL at PMN concentrations greater than 0.5 × 10^6^ PMN/mL (*p* < 0.001).

**Figure 5 pharmaceuticals-16-01501-f005:**
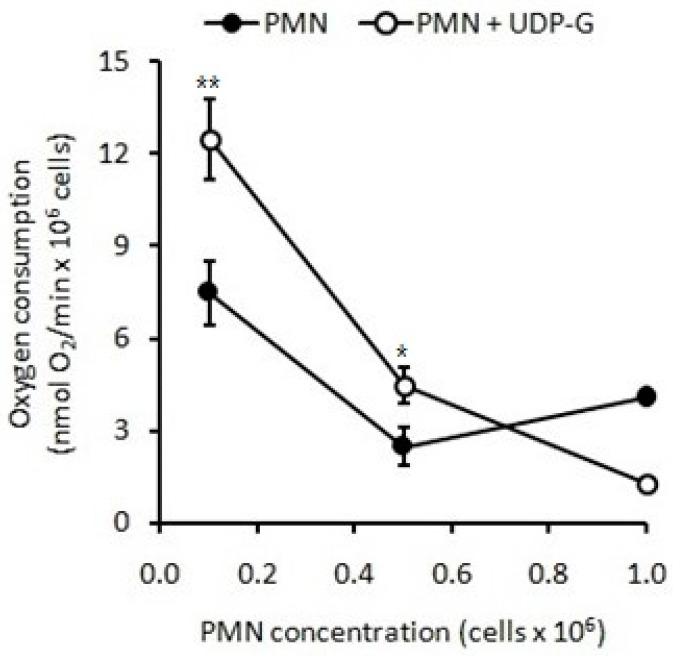
UDP-G increased O_2_ consumption at low PMN concentrations (** *p* < 0.001; * *p* < 0.01). Inversely, a decreased O_2_ consumption is observed at PMN concentrations higher than 0.8 × 10^6^ PMN/mL (*p* < 0.001).

**Figure 6 pharmaceuticals-16-01501-f006:**
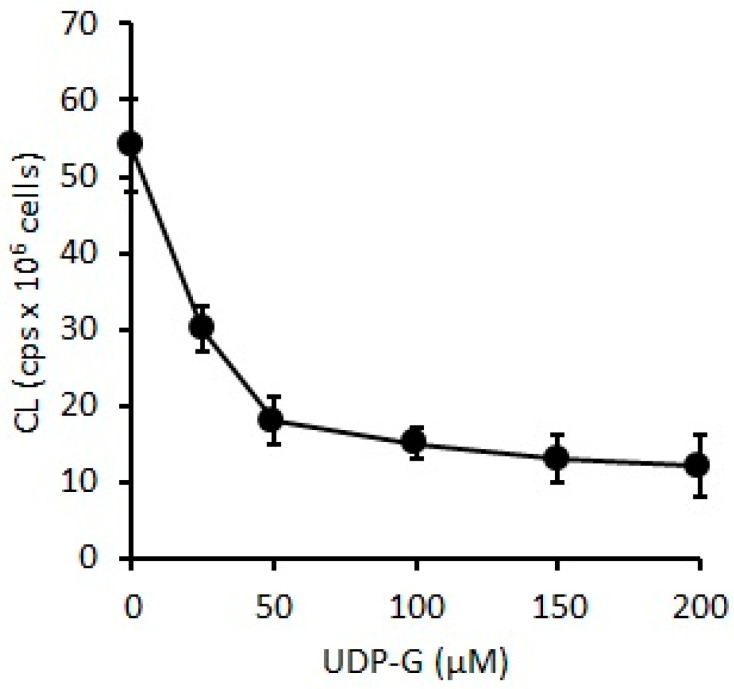
Changes in OS levels measured as CL in PMN cells (0.1 × 10^6^ PMN cells/mL) incubated with increasing UDP-G concentrations (25 µM, *p* < 0.05 and 50–150 µM, *p* < 0.001).

**Figure 7 pharmaceuticals-16-01501-f007:**
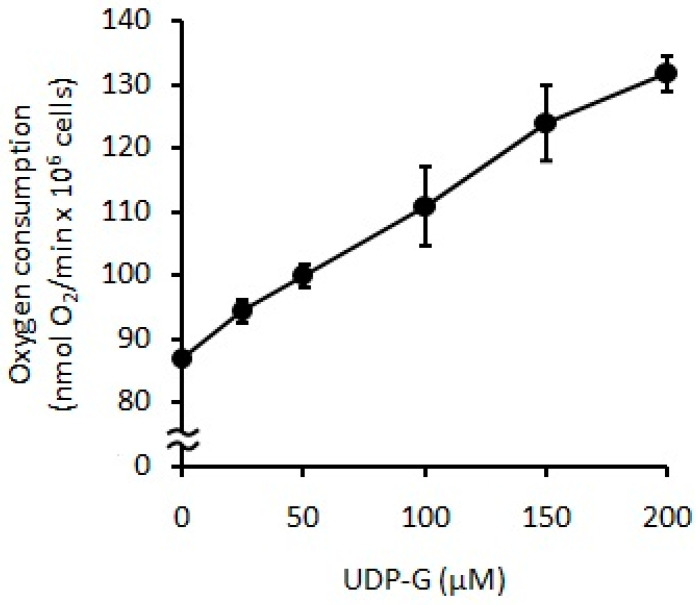
Changes in RB measured as oxygen consumption in PMN cells (0.1 × 10^6^ PMN cells/mL) incubated with increasing concentrations of UDP-G (*p* < 0.0001).

**Table 1 pharmaceuticals-16-01501-t001:** Oxygen consumption of 0.5 × 10^6^ PMN/mL after treatment with saline solution (PBS) (Control), LPS, or PMA.

PMN + Agonist	Oxygen Consumption (nmol O_2_/min/10^6^ PMN)
Control (PBS)	1.8 ± 1.2
LPS	2.1 ± 0.8 ***
PMA	2.8 ± 0.6 *

* *p* < 0.05; *** *p* < 0.001.

**Table 2 pharmaceuticals-16-01501-t002:** Increase in RB in PMNs induced by LPS, PMA, or UDP-G at concentrations of 0.1 × 10^6^ PMN/mL and 0.5 × 10^6^ PMN/mL.

PMN + Agonist	Oxygen Consumption (nmol O_2_/min/10^6^ PMN)	
	0.1 × 10^6^ PMN/ mL	0.5 × 10^6^ PMN/ mL
Control (PBS)	7.2 ± 1.3	1.8 ± 1.2
LPS	7.5 ± 2.0	2.1 ± 0.7
PMA	10 ± 2 ***	2.5 ± 1.5 *
UDP-G	13 ± 1.5 ***	4.0 ± 1.0 **

** p* < 0.05; ** *p* < 0.01; *** *p* < 0.001.

**Table 3 pharmaceuticals-16-01501-t003:** Oxygen consumption in PMNs treated with LPS, PMA, UDP-G. or LPS + UDP-G at very low concentrations of PMNs (0.02 × 10^6^ PMN/mL).

PMN + Agonist	Oxygen Consumption (nmol O_2_/min/10^6^ PMN)
	0.02 × 10^6^ PMN/ mL
Control (PBS)	55 ± 10
LPS	60 ± 5
PMA	52 ± 1
UDP-G	60 ± 5
LPS + UDP-G	69 ± 4 *

* *p* < 0.05.

**Table 4 pharmaceuticals-16-01501-t004:** Stimulation index of CL and RB for LPS, PMA, and UDP-G of PMNs (0.1 × 10^6^ PMN/mL).

Agonist	CL-SI	RB-SI
None	1.0	1.0
LPS	1.7	1.2
PMA	2.2	1.4
UDP-G	1.6	1.8

## Data Availability

Data is contained within the article.

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
