# Peer review of "Uridine Diphosphate Glucose (UDP-G) Activates Oxidative Stress and Respiratory Burst in Isolated Neutrophils"

_pharmaceuticals, 2023, doi:10.3390/ph16101501_

Round 1

Reviewer 1 Report

Manuscript title: Uridine-Diphosphate-Glucose (UDP-G) activates oxidative stress and respiratory burst in isolated neutrophils

Manuscript ID: Pharmaceuticals-2637613

. . . . . . . . . . . . . . . . . . . . . . . . . . . . . . . . . . . . . . . . . . . . . . . . . . . . . . . . . . . . . . . . . . . . . . . . . . . . . . . . . . . . .

Comments on this research paper on Uridine-Diphosphate-Glucose activation of oxidative stress and respiratory burst in isolated neutrophils are presented below.

It would be more appropriate to italicize some technical terms such as "in vitro" and "ex vivo" in the text of the manuscript.

It would be appropriate to use some abbreviations together with their clear expression in the first place where they are used, and then to include only the abbreviation.

Shouldn't it be written "eustress" instead of "eutress"?

“After collection, the sample was allowed to clot undisturbed for 30 minutes.” Please check the accuracy of the above sentence. Because this study was performed on whole blood samples anticoagulated with EDTA.

Please check the phrase "MPA stimulation".

To determine the effect of UDP-G on PMN function, increasing concentrations of UDP- G (0-200 µM) were used. On what basis was this dose range determined?

Figure 1, 2 and 7, 8 in the manuscript is not very clear, so it would be useful to compensate for this with a clearer figure.

How were the doses of LPS and PMA applied to PMNs determined?

In the discussion section of the article, authors should make a clear consideration of the issue to which their study findings are directly relevant.

. . . . . . . . . . . . . . . . . . . . . . . . . . . . . . . . . . . . . . . . . . . . . . . . . . . . . . . . . . . . . . . . . . . . . . . . . . . . . . . . . . . . .

Author Response

In this new version of the manuscript you can find in the highlighted document, in red the words, sentences and paragraphs that have been removed from the the text and in yellow highlights the new words, sentences, paragraphs and changes made.

                We responded to your questions and comments and incorporate them into the text, taking into account your valuable comments, we believe that including them and the corresponding responses to the revised version enhance the article, improving the quality of the analysis of the results. For this reason we have incorporated several paragraphs and sentences in the text highlighted in yellow. We appreciate your comments as they increase the quality of our work. In all cases we indicate the specific page where the modifications have occurred and were included in the manuscript (page numbers correspond to the highlighted version of the manuscript).

                Some references were deleted in the manuscript, in text and list of references. For this reason references in text and list were reorganized.

Answers to the specific comments of the Reviewer #1:

Comments on this research paper on Uridine-Diphosphate-Glucose activation of oxidative stress and respiratory burst in isolated neutrophils are presented below.

  • Reviewer: It would be more appropriate to italicize some technical terms such as "in vitro" and "ex vivo" in the text of the manuscript.

Response: The term "in vitro" has been italicized in the text of the manuscript.              The term “ex vivo” was removed because the sentence “Since this is the first study using spontaneous chemiluminescence in ex vivo living PMN cells in suspension, to” was replaced by “In order to” (page 3).

  • Reviewer: It would be appropriate to use some abbreviations together with their clear expression in the first place where they are used, and then to include only the abbreviation.
    Response: Some abbreviations were included in the text (CL, OS, RB), and the clear expression was included only in the first place in the text where they are used. A list of abbreviations has been included at the end of the conclusions, as suggested by Reviewer #3 (page 14).

  • Reviewer: Shouldn't it be written "eustress" instead of "eutress"?
    Response: Thank you for noticing this. The term has been corrected (pages 2 and 13).

  • Reviewer: “After collection, the sample was allowed to clot undisturbed for 30 minutes.” Please check the accuracy of the above sentence. Because this study was performed on whole blood samples anticoagulated with EDTA.

Response: Thank you for noticing this mistake. This sentence was eliminated because plasma was used for experiments (page 12).

  • Reviewer: Please check the phrase "MPA stimulation".
    Response: The following phrase “The standardization of optimal PMN concentration for the oxidative stress evaluation was developed by the CL tested at different cell concentrations, both under control conditions (spontaneous CL) as well as with PLS and MPA stimulation” was removed and replaced with “the optimal PMN concentration required to evaluate the oxidative stress in PMN cells was carried out by assessing the CL achieved at different cell concentrations, in non-stimulated cells and in the presence of LPS or PMA” (page 13).

  • Reviewer: To determine the effect of UDP-G on PMN function, increasing concentrations of UDP- G (0-200 µM) were used. On what basis was this dose range determined?

Response: This dose range was used because at a concentration of 100 µM UDP- G was reported to promote QTX and inflammatory response [16,17], lower and higher than 100 µM concentration were evaluated in this research for the evaluation of oxidative stress effect on PMNs (Materials and methods, pages 12 and 13).

  • Reviewer: Figure 1, 2 and 7, 8 in the manuscript is not very clear, so it would be useful to compensate for this with a clearer figure.

Response: Thank you for your comment. Figures 1 and 2 were replaced with new ones in the text. Figures 7 and 8 has been replaced by Tables 2 and 3 respectively, as was suggested by Reviewer #2.

  • Reviewer: How were the doses of LPS and PMA applied to PMNs determined?
    Response: The concentrations of these stimulating agents of the respiratory burst have been employed based on previous experience from our lab. The concentrations of 20 ng/mL PMA and 2 ug/mL LPS are also concentrations that are regularly used to trigger oxidative burst in these cells. This explanation was added in the text: “These LPS and PMA concentrations are regularly used to trigger oxidative burst in PMNs [20], 100 µM of UDP-G was reported to promote QTX and inflammatory response [16,17]” (Materials and methods, page 12)

  • Reviewer: In the discussion section of the article, authors should make a clear consideration of the issue to which their study findings are directly relevant.

Response:  Some paragraphs were removed, in page 9: “Several studies have demonstrated that UDP-sugars can be released in a regulated manner from several cell types, including from cells of airway epithelial. Interestingly, in primary cultures of human bronchial epithelial cells infected with respiratory syncytial virus or treatment with interleukin 13 (IL-13), as well as goblet cell-like Calu-3 cells, exhibited an enhanced release of UDP-G concomitantly with increased mucin secretion after infection with respiratory syncytial virus [29,30]”  and in page 10: “More recently, using an experimental model of ischemic acute kidney injury (AKI), it was demonstrated that either blockade of the proinflammatory P2Y14 receptor located on the apical membrane of intercalated cells (ICs) or ablation of the gene encoding the P2Y14 receptor in ICs, inhibited IRI-induced increase of chemokine expression in ICs, reduced neutrophil and monocyte renal infiltration, reduced the extent of kidney dysfunction, and attenuated proximal tubule damage. Furthermore in humans, the concentration of UDP-G was higher in urine samples from intensive care unit patients who developed AKI compared with patients without AKI [36].All these evidences not only indicate that P2Y14-R activation participate in the inflammatory tissue damage, but also high concentrations of UDP-G could play a role in the development of this damage, inducing PMN accumulation in the tissular ischemic area [37], as previously demonstrated in lung and kidney [38]” and in pages 10 and 11: “In this sense, Andrew E Williams and Rachel C Chambers hypothesize that hypoxia could plays a central role in the degranulation of PMNs [42], a mechanism directly related with NETs formation, because granule-derived proteins are required for the major neutrophil functions, such as chemotaxis, antimicrobial function and NET release [43,44]. It was demonstrated that under hypoxia, ROS are reduced due to the lack of available molecular oxygen [45]. To date, it is not known with precision whether hypoxia influences the release of UDP-G by the bronchial epithelium, nor what is the role of extracellular UDP-G in the degranulation of PMNs and/or the formation of NETs in hypoxic conditions”.

Reference numbers were changed in the text.

The discussion includes the paragraphs considering the most relevant findings of the study and the practical aspects of the observed results. An initial paragraph indicating the strengths and limitations of the study was incorporated (page 9): “This is the first study that allows the evaluation of the inflammatory response and OS of living cells, ex vivo, by measuring in PMNs the oxygen consumption and CL. This research provides optimal working conditions for the determination of OS in these cells using sensitive techniques. This work shows that this methodology can be applied to evaluate the functionality of living cells in physiological and/or pathological conditions, as well as the effect of stimulating or inhibitory agents of the inflammatory, oxidant or antioxidant response. In this study, it was observed that at the extracellular level UDP-G exerts a dual effect on exposed PMNs, an increase in RB, ROS production, and oxidative damage to membrane lipids at lower PMN and UDP-G concentrations. However, UDP-G concentrations greater than those that promote chemotaxis and inflammatory response (100 uM) increase the RB but decrease oxidative damage, indicating that the ROS generated are destined for the inflammatory and defense response of the PMNs. However, the main limitations of this study are that the cells are required to be kept alive throughout the experiment, they have to be isolated from the blood on the day of sampling, and both oxygen consumption and CL must be measured immediately”).

Reviewer 2 Report

The manuscript entitled „Uridine-Diphosphate-Glucose (UDP-G) activates oxidative stress and respiratory burst in isolated neutrophils” presents interesting issue, but some problems should be corrected.

General:

Authors have in their manuscript some corrections in red (e.g. Abstract) – the highlights should be removed.

The English language needs to be polished within the whole manuscript, e.g. instead of “Changes oxidative stress”, it should be “Changes of oxidative stress”, or “Changes in oxidative stress” – it is only one example, but the corrections are needed within the whole manuscript.

Abstract:

Authors should try to present within their Abstract the most important information, including the practical aspects of their results (even if Abstract needs to be a little bit longer)

Introduction:

Authors should remove the redundant information and focus on the most important to justify the need for conducting their study.

Authors should not focus on their own studies (“We have previously been able to demonstrate…”) and even if they present the results of their own study, they do not have to indicate it or they should do it n more objective way (e.g. “In the previous own study it was indicated that…”).

Results:

Figure 1, Figure 2, Figure 7, Figure 8 – the quality of the figures is very poor and it should be corrected

Some figures should be replaced by tables to be easier to follow: Figure 5 Figure 7, Figure 8 (to present exact numeric values)

In each case when Authors indicate “changes” the adequate p-Value should be presented to indicate that the changes were statistically significant.

Discussion:

The practical aspects of the observed results should be addressed within this section.

Authors should present in details strengths and limitations of their study.

Materials and methods:

The statistical analysis is not addressed adequately

The applied software should be indicated

It seems that Authors did not verify the normality of distribution of their data – they should do it and present the related methodology.

After verifying the normality of distribution, in case of parametric distribution mean ± SD should be presented, while for nonparametric distribution – median accompanied by minimum and maximum value.

The SEM is not properly applied, as it measures rather a precision for the estimated population mean and does not present the variability of data around the mean (while SD does), so instead of SEM, a SD should be applied.

Author Contribution:

It seems that contribution of some of Authors was only minor and they did not participate in preparing manuscript. There is a serious risk of a guest authorship procedure which is forbidden. In such case (if they did not participate in manuscript preparation) they should be rather presented in Acknowledgements Section and not be indicated as authors of the study.

The English language needs to be polished within the whole manuscript, e.g. instead of “Changes oxidative stress”, it should be “Changes of oxidative stress”, or “Changes in oxidative stress” – it is only one example, but the corrections are needed within the whole manuscript.

Author Response

In this new version of the manuscript you can find in the highlighted document, in red the words, sentences and paragraphs that have been removed from the the text and in yellow highlights the new words, sentences, paragraphs and changes made.

                We responded to your questions and comments and incorporate them into the text, taking into account your valuable comments, we believe that including them and the corresponding responses to the revised version enhance the article, improving the quality of the analysis of the results. For this reason we have incorporated several paragraphs and sentences in the text highlighted in yellow. We appreciate your comments as they increase the quality of our work. In all cases we indicate the specific page where the modifications have occurred and were included in the manuscript (page numbers correspond to the highlighted version of the manuscript).

                Some references were deleted in the manuscript, in text and list of references. For this reason references in text and list were reorganized.

Answers to the specific comments of the Reviewer #2:

The manuscript entitled “Uridine-Diphosphate-Glucose (UDP-G) activates oxidative stress and respiratory burst in isolated neutrophils” presents interesting issue, but some problems should be corrected.

General:

  • Reviewer: Authors have in their manuscript some corrections in red (e.g. Abstract) – the highlights should be removed.

Response: Thank you for noticing this. The corrections in red and highlights have been removed (page 1, Abstract).

  • Reviewer: The English language needs to be polished within the whole manuscript, e.g. instead of “Changes oxidative stress”, it should be “Changes of oxidative stress”, or “Changes in oxidative stress” – it is only one example, but the corrections are needed within the whole manuscript.
    Response: Thank you for this observation. The phrase “Changes oxidative stress” has been removed and replaced with “Changes in oxidative stress” (page 8). The English language was improved by a native speaker and was polished within the whole manuscript.

  • Reviewer: Abstract: Authors should try to present within their Abstract the most important information, including the practical aspects of their results (even if Abstract needs to be a little bit longer).

Response: The abstract has been reformed to better reflect the main findings of the study.

  • Reviewer: Introduction: Authors should remove the redundant information and focus on the most important to justify the need for conducting their study.

Response: Redundant information was removed from the introduction of the manuscript: page 2: “its predecessor molecules which are formed in the phospholipid peroxidation chain reaction”; page 3: “due to the final stage of the phospholipids oxidation in chain reaction”.

  • Reviewer: Authors should not focus on their own studies (“We have previously been able to demonstrate…”) and even if they present the results of their own study, they do not have to indicate it or they should do it n more objective way (e.g. “In the previous own study it was indicated that…”).

Response: Thank you for noticing this. The phrase “we have previously been able to demonstrate” has been removed and replaced with “In previous work, it was been shown that” (page 2)

  • Reviewer: Results: Figure 1, Figure 2, Figure 7, Figure 8 – the quality of the figures is very poor and it should be corrected.

Response: Thank you for your comment. Figures 1 and 2 were replaced with new ones in the text, and Figures 5, 7 and 8 were replaced by tables 1, 2 and 3 respectively, as was suggested by Reviewer #2 in the following item 7.

  • Reviewer: Some figures should be replaced by tables to be easier to follow: Figure 5 Figure 7, Figure 8 (to present exact numeric values)

Response: Figures 5, 7 and 8 were replaced by tables (Tables 1, 2 and 3, respectively).

  • Reviewer: In each case when Authors indicate “changes” the adequate p-Value should be presented to indicate that the changes were statistically significant.

Response: The adequate p-value were added to indicate that the changes were significant (for increases or decreases), both in the texts and in the figure legends (pages 3 to 9).

  • Reviewer: Discussion: The practical aspects of the observed results should be addressed within this section.

Response: Some redundant paragraphs were removed from the discussion (page 9): “Several studies have demonstrated that UDP-sugars can be released in a regulated maner from several cell types, including from cells of airway epithelial. Interestingly, in primary cultures of human bronchial epithelial cells infected with respiratory syncytial virus or treatment with interleukin 13 (IL-13), as well as goblet cell-like Calu-3 cells, exhibited an enhanced release of UDP-G concomitantly with increased mucin secretion after infection with respiratory syncytial virus [29,30]”); and page 10: “More recently, using an experimental model of ischemic acute kidney injury (AKI), it was demonstrated that either blockade of the proinflammatory P2Y14 receptor located on the apical membrane of intercalated cells (ICs) or ablation of the gene encoding the P2Y14 receptor in ICs, inhibited IRI-induced increase of chemokine expression in ICs, reduced neutrophil and monocyte renal infiltration, reduced the extent of kidney dysfunction, and attenuated proximal tubule damage. Furthermore in humans, the concentration of UDP-G was higher in urine samples from intensive care unit patients who developed AKI compared with patients without AKI [36]. All these evidences not only indicate that P2Y14-R activation participate in the inflammatory tissue damage, but also high concentrations of UDP-G could play a role in the development of this damage, inducing PMN accumulation in the tissular ischemic area [37], as previously demonstrated in lung and kidney [38]” and in pages 10 and 11: “In this sense, Andrew E Williams and Rachel C Chambers hypothesize that hypoxia could plays a central role in the degranulation of PMNs [42], a mechanism directly related with NETs formation, because granule-derived proteins are required for the major neutrophil functions, such as chemotaxis, antimicrobial function and NET release [43,44]. It was demonstrated that under hypoxia, ROS are reduced due to the lack of available molecular oxygen [45]. To date, it is not known with precision whether hypoxia influences the release of UDP-G by the bronchial epithelium, nor what is the role of extracellular UDP-G in the degranulation of PMNs and/or the formation of NETs in hypoxic conditions”.

The discussion includes the paragraphs considering the most relevant findings of the study and the practical aspects of the observed results. An initial paragraph with strengths and limitations of the study (page 9) was incorporated. Reference numbers were changed in the text.

  • Reviewer: Authors should present in details strengths and limitations of their study.

Response: Thank you for your comment. The inclusion of the strengths and limitations of the study allows us to enrich the discussion of the results in the manuscript. The strengths and limitations of the study were included in the manuscript (page 9). Strenghts: This is the first study that allows the evaluation of the inflammatory response and oxidative stress of living cells, ex vivo, by measuring the oxygen consumption and chemiluminescence. This research provides optimal working conditions for the determination of oxidative stress in these cells using sensitive techniques. This work shows that this methodology can be applied to evaluate the functionality of living cells in physiological and/or pathological conditions, as well as the effect of stimulating or inhibitory agents of the inflammatory, oxidant or antioxidant response. In this study, it was observed that at the extracellular level UDP-G exerts a dual effect on exposed PMNs, an increase in respiraory burst, ROS production, and oxidative damage to membrane lipids at lower PMN and UDP-G concentrations. However, UDP-G concentrations greater than those that promote chemotaxis and inflammatory response (100 uM) increase the respiratory burst but decrease oxidative damage, indicating that the reactive species generated are destined for the inflammatory and defense response of the PMNs. Limitations: The main limitations of this study are that the cells are required to be kept alive throughout the experiment, they have to be isolated from the blood on the day of sampling, and both oxygen consumption and chemiluminescence must be measured immediately.

  • Materials and methods:

Reviewer: The statistical analysis is not addressed adequately.

Response: The statistical analysis has been addressed adequately in the text, in material and methods section of the manuscript and all the results showed in text, tables and figures.

  • Reviewer: The applied software should be indicated.

Response: The software used for statistical analysis was GraphPad InStat3, and GraphPad Prism4 software (GraphPad Software Inc., San Diego, California, USA) (page 13).

  • Reviewer: It seems that Authors did not verify the normality of distribution of their data – they should do it and present the related methodology.

Response: The normality of distribution of data was verified, and was presented in methodology (page 13). Results are expressed as mean ± SD for normally distributed data. Kolmogorov Smirnov test was performed to evaluate variables distribution. Differences between groups were tested using unpaired Student’s t-test, one-way Analysis of Variances (ANOVA), followed by the parametric Dunnet post hoc test. The software GraphPad InStat3 and GraphPad Prism4 software (GraphPad Software Inc., San Diego, California, USA) were used. Significance between groups differences are indicated by p value, where p<0.05 was considered significant, p<0.01 was considered very significant and p<0.001 was considered extremely significant (page 13).

  • Reviewer: After verifying the normality of distribution, in case of parametric distribution mean ± SD should be presented, while for nonparametric distribution – median accompanied by minimum and maximum value.

Response: Normality of distribution was verifying and parametric test was used. Data are presented as mean ± SD.

  • Reviewer: The SEM is not properly applied, as it measures rather a precision for the estimated population mean and does not present the variability of data around the mean (while SD does), so instead of SEM, a SD should be applied.

Response: Thank you for your comment. SD was applied in data, figures and tables. Statistical analysis has been improved.

  • Reviewer: Author Contribution: It seems that contribution of some of Authors was only minor and they did not participate in preparing manuscript. There is a serious risk of a guest authorship procedure which is forbidden. In such case (if they did not participate in manuscript preparation) they should be rather presented in Acknowledgements Section and not be indicated as authors of the study.

Response: All authors have read and participated in the preparation of the manuscript, however in the text only the main activity of each author was indicated. All the activities of each author were incorporated into the corrected version of the manuscript (page 14).

Reviewer 3 Report

Author proposed a paper entitled “Uridine-Diphosphate-Glucose (UDP-G) activates oxidative stress and respiratory burst in isolated neutrophils” for the publication in Pharmaceuticals, MDPI.

This paper has a good scientific soundness and deserves to be published after performing some revisions.

PMN, DP-G, OS, RB, CL, LPS, PMA, SI, QTX, ETDA, etc could be included in an abbreviation list, according to the guidelines of this Journal.

The quality of English should be revised accordingly.

Here is the list of my issues, point by point:

line 37. Check syntax here.

Line 42. A reference is needed here.

Line 65 “may be distinguish” should be “distinguished”?

Line 84. “will depend on the” future tense is not necessary in this context.

Line 89. “most abundant circulating white blood cells in humans,” ok; however, it could be better to quantify this abundance in terms of percentage, if possible.

Iine 98. Please add a reference here.

I would rephrase lines from 101 to 104 in order to clearly define what is the scope of this paper.

Line 107. “ex vivo” should be written in italics.

Figure 1. Please improve the figure and clarify the necessity of red dots too link figure 1a and figure 1b.

In the same manner, I would improve figure 2, increasing the focus and the quality as for the following singular figures (from figure 3 to 6)

Same problem in figure 7.

Line 235. “and induce the release the potent neutrophil chemoattractant” please check the syntax here.

Line 237. “in vitro” should be in italics.

Line 280 “It is need to better under” should be checked and modified as “It is needed…”

Conclusions: please give some ideas about your intentions for the future.

Use of English deserves a proper moderate revision.

Author Response

In this new version of the manuscript you can find in the highlighted document, in red the words, sentences and paragraphs that have been removed from the the text and in yellow highlights the new words, sentences, paragraphs and changes made.

                We responded to your questions and comments and incorporate them into the text, taking into account your valuable comments, we believe that including them and the corresponding responses to the revised version enhance the article, improving the quality of the analysis of the results. For this reason we have incorporated several paragraphs and sentences in the text highlighted in yellow. We appreciate your comments as they increase the quality of our work. In all cases we indicate the specific page where the modifications have occurred and were included in the manuscript (page numbers correspond to the highlighted version of the manuscript).

                Some references were deleted in the manuscript, in text and list of references. For this reason references in text and list were reorganized.

Answers to the specific comments of the Reviewer #3:

Author proposed a paper entitled “Uridine-Diphosphate-Glucose (UDP-G) activates oxidative stress and respiratory burst in isolated neutrophils” for the publication in Pharmaceuticals, MDPI.

This paper has a good scientific soundness and deserves to be published after performing some revisions.

  • Reviewer: PMN, DP-G, OS, RB, CL, LPS, PMA, SI, QTX, ETDA, etc could be included in an abbreviation list, according to the guidelines of this Journal.

Response: The following list of abbreviations has been included (page 14):

CL: Chemiluminescence

EDTA: Ethylenediaminotetraacetic acid

LPS: Lipopolysaccharide

OS: Oxidative stress

PMA: Phorbol 12-myristate 13-acetate

PMN: polymorphonuclear neutrophil

QTX: Chemotaxis

RB: Respiratory Burst

SI: Stimulation Index

UDP-G: Uridine-Diphosphate-Glucose

  • Reviewer: The quality of English should be revised accordingly.

Response: The quality of the paper has been  improved by a native speaker.

Reviewer: Here is the list of my issues, point by point:

  • Reviewer: line 37. Check syntax here.

Response: The line “UDP-G increased respiratory burst and NADPH oxidase complex in a dose-dependent manner” has been replaced with “UDP-G triggered the respiratory burst and NADPH oxidase activity in a dose-dependent manner.”(page 11).

  • Reviewer: Line 42. A reference is needed here.

Response: The reference 20 was included (page 2).

  • Reviewer: Line 65 “may be distinguish” should be “distinguished”?

Response: the phrase may be distinguish has been replaced by “should be distinguished “on line 65 (page 2).

  • Reviewer: Line 84. “will depend on the” future tense is not necessary in this context.

Response: The phrase “will depend on the” has been  replaced  by “is dependent on” (page 2).

  • Reviewer: Line 89. “most abundant circulating white blood cells in humans,” ok; however, it could be better to quantify this abundance in terms of percentage, if possible.

Response: The relative abundance of PMNs has been specified in the following phrase: (typically between 40-75%) (page 3)

  • Reviewer: Line 98. Please add a reference here.
    Response: The reference 25 was added (page 3).

  • Reviewer: I would rephrase lines from 101 to 104 in order to clearly define what is the scope of this paper.

Response: The scope of the article suggests a key role for UDP-G in the PMN response to infection and/or sepsis (page 3). The scope and objective of the research were added in the introduction (page 3): The objective of this research was to demonstrate if one of the biochemical mechanisms involved in the activation of PMNs by UDP-G is OS.

  • Reviewer: Line 107. “ex vivo” should be written in italics.

Response: ex vivo has been written in italics (page 4), but this term and the corresponding sentence where it is insert, was removed for improving the quality of the manuscript.

  • Reviewer: Figure 1. Please improve the figure and clarify the necessity of red dots too link figure 1a and figure 1b.

Response: Figure 1 has been improved. Figure 1a was removed and  only Figure 1b is shown  in the modified version of the manuscript, because this is the figure  relevant for the manuscript. Figure 1 was replaced for the new one.

  • Reviewer: In the same manner, I would improve figure 2, increasing the focus and the quality as for the following singular figures (from figure 3 to 6)

Response: Figure 2 has been  improved. Figures 2a and 2c were removed and only Figure 2b is shown in the modified version of the manuscript, because this is the figure relevant for the manuscript. Figure 2 was replaced for the new one.

  • Reviewer: Same problem in figure 7.

Response: Figure 7 was  replaced by table 2 as suggested by Reviewer 2#.

  • Reviewer: Line 235. “and induce the release the potent neutrophil chemoattractant” please check the syntax here.

Response: The syntax has been reviewed and the phrase “induce the release the potent neutrophil chemoattractant interleukin 8” has been replaced by “induce the release of the potent neutrophil chemoattractant interleukin 8 (Il-8)” (page 9).

  • Reviewer: Line 237. “in vitro” should be in italics.

Response: The phrase has been rewritten in italics (page 10).

  • Reviewer: Line 280 “It is need to better under” should be checked and modified as “It is needed…”

Response: The phrase it is need to better under-standing” has been replaced with “it is needed to better under-standing” (page 10).

  • Reviewer: Conclusions: please give some ideas about your intentions for the future.
    Response: The following sentence was added in the conclusion: As a future perspective, based on these results we hope to advance with the knowledge of the induction of Hepcidin synthesis and investigate whether UDP-G plays a role in the regulation of the loading of Hepcidin transcripts (a protein that regulates iron metabolism, which increases in patients with chronic clinical conditions such as sepsis, myelomas, arthritis, burns and inflammatory bowel disease), not in the liver but in activated PMNs, in response to proinflammatory stimuli in addition to the increase in respiratory burst and oxidative stress (page 14).

  • Reviewer: Comments on the Quality of English Language. Use of English deserves a proper moderate revision.

Response: The quality of the paper has been improved by a native English speaker.

Round 2

Reviewer 1 Report

Manuscript title: Uridine-Diphosphate-Glucose (UDP-G) activates oxidative stress and respiratory burst in isolated neutrophils

Manuscript ID: Pharmaceuticals-2637613 R2

. . . . . . . . . . . . . . . . . . . . . . . . . . . . . . . . . . . . . . . . . . . . . . . . . . . . . . . . . . . . . . . . . . . . . . . . . . . . . . . . . . . . .

Comments on this research paper on Uridine-Diphosphate-Glucose activation of oxidative stress and respiratory burst in isolated neutrophils are presented below.

It was determined that the authors made the necessary corrections in line with the referees' opinions.

and but the last paragraph added at the end of the article is, in my opinion, unnecessary and incompatible with the integrity of the subject and should be removed.

“As a future perspective, based on these results we hope to advance with the  knowledge of the induction of Hepcidin synthesis and investigate whether UDP-G plays a role in the regulation of the loading of Hepcidin transcripts (a protein that regulates iron metabolism, which increases in patients with chronic clinical conditions such as sepsis, myelomas, arthritis, burns and inflammatory bowel disease), not in the liver but in activated PMNs, in response to proinflammatory stimuli in addition to the increase in respiratory burst and oxidative stress.”

. . . . . . . . . . . . . . . . . . . . . . . . . . . . . . . . . . . . . . . . . . . . . . . . . . . . . . . . . . . . . . . . . . . . . . . . . . . . . . . . . . . . .

Author Response

Dear Reviewer #1: 

We greatly appreciate your comment, the paragraph "As a future perspective, based on these results we hope to advance with the knowledge of the induction of Hepcidin synthesis and investigate whether UDP-G plays a role in the regulation of the loading of Hepcidin transcripts (a protein that regulates iron metabolism, which increases in patients with chronic clinical conditions such as sepsis, myelomas, arthritis, burns and inflammatory bowel disease), not in the liver but in activated PMNs, in response to proinflammatory stimuli in addition to the increase in respiratory burst and oxidative stress” was removed from the manuscript

Reviewer 3 Report

Authors responded point by point to my issues. The use of English has improved through an overall revision of the manuscript. Now the paper deserves to be published, in my opinion.

Improvement of the use of English

Author Response

Dear Reviewer #3, 

Thank you very much for your comment.